# Effects of Replacing Soybean Meal with Enzymatically Fermented Citric Waste Pellets on In Vitro Rumen Fermentation, Degradability, and Gas Production Kinetics

**DOI:** 10.3390/ani15162351

**Published:** 2025-08-11

**Authors:** Gamonmas Dagaew, Seangla Cheas, Chanon Suntara, Chanadol Supapong, Anusorn Cherdthong

**Affiliations:** 1Tropical Feed Resources Research and Development Center (TROFREC), Department of Animal Science, Faculty of Agriculture, Khon Kaen University, Khon Kaen 40002, Thailand; gamonmasdagaew@gmail.com (G.D.); seangla.c@kkumail.com (S.C.); chansun@kku.ac.th (C.S.); 2Department of Animal Science, Faculty of Agricultural Innovation and Technology, Rajamangala University of Technology Isan, Nakhon Ratchasima 30000, Thailand; chanadol@kkumail.com

**Keywords:** feed additives, microbiota, methane mitigation, rumen dynamic, ruminant nutrition

## Abstract

Soybean meal (SBM) is a widely used protein source in ruminant diets, but its high cost and dependence on imports pose challenges, especially in developing countries. This study investigated whether citric waste fermented with yeast waste with enzymes in pellet form (CWYWEP), produced from agro-industrial by-products such as citric and yeast waste, could fully replace SBM under in vitro conditions. The results demonstrated that a 100% replacement with CWYWEP improved gas production, nutrient degradability, and fermentation efficiency, while also reducing methane (CH_4_) emissions. Unlike non-enzymatic alternatives, CWYWEP offered a high-protein, digestible, and environmentally friendly feed option without adversely affecting rumen pH or microbial populations. These findings highlight CWYWEP as a promising and sustainable alternative to completely replace SBM in ruminant feeding systems.

## 1. Introduction

The global livestock sector is increasingly challenged by the high cost and limited availability of conventional protein sources, particularly SBM, which remains a principal component in ruminant diets. While SBM provides high-quality protein, its reliance on imported sources and escalating prices contributes substantially to production costs, especially in developing countries [1]. In response, the search for alternative, locally available, and sustainable protein feedstuffs has intensified, with an emphasis on maintaining animal performance while reducing environmental and economic burdens [2]. This shift aligns with the principles of the Bio-Circular-Green (BCG) economic model, which promotes the valorization of agro-industrial by-products to support circular feed systems, reduce waste, and enhance feed resource sustainability [3].

Citric waste (CW), a by-product of citric acid production, is the second most abundant fermentation by-product globally, with an estimated annual output of 1.7 million tons [4]. While CW is rich in pectin, soluble fiber, and bioactive compounds [5], its high fiber content limits its direct use in ruminant feeding due to poor digestibility. Previous studies have shown that fermenting CW with yeast waste (YW)—a nitrogen-rich by-product from bioethanol production—can enhance its crude protein (CP) content significantly, reaching up to 53.5% on a DM basis [6]. However, despite its improved nutrient profile, fermented CWYW remains bulky, highly perishable, and nutritionally inconsistent, making it difficult to store and handle under practical farm conditions.

Pelleting is a widely adopted feed-processing method that improves feed density, transportability, storage stability, and intake consistency in ruminant diets [7]. However, pelleted CWYW (CWYWP) still contains a substantial proportion of structural carbohydrates, which may restrict nutrient availability and digestibility. The use of exogenous fibrolytic enzymes—such as cellulase, xylanase, and mannanase—has been shown to enhance the degradation of fiber components, thereby improving the digestibility of DM and neutral detergent fiber (NDF) by facilitating microbial colonization and fermentation in the rumen [8]. Incorporating such enzyme complexes during fermentation may further optimize the nutritional quality and fermentative value of high-fiber by-product feeds [9].

To address the limitations of high fiber and poor digestibility in CWYWP, this study introduces a novel approach by incorporating a multi-enzyme complex—comprising cellulase, xylanase, and mannanase—during the fermentation process with yeast waste [5,7]. These enzymes are known to hydrolyze structural polysaccharides into fermentable sugars, enhancing substrate availability for rumen microbes [8]. In parallel, yeast waste provides readily available nitrogen sources, which may synergize with enzymes to stimulate microbial growth and enzymatic activity. The combined action is expected to improve nutrient digestibility, support early microbial colonization, and increase volatile fatty acid (VFA) production while potentially mitigating CH_4_ output [4,5]. This mechanism offers a strategic advantage over conventional CWYWP by enhancing both protein enrichment and fiber breakdown during the fermentation process.

Despite these advances, limited research has been conducted on citric waste fermented with both yeast waste and exogenous enzymes, subsequently pelleted into a product designed for ruminant feeding. While studies using yeast-fermented CW without enzyme addition have reported modest improvements in protein content and fermentation profiles [10], none have explored the combined use of yeast and fibrolytic enzymes during fermentation, followed by pelleting, as a viable protein-rich replacement for SBM. This unexplored approach represents a promising yet under-investigated strategy to simultaneously enhance digestibility and feed quality using agro-industrial by-products [7].

Building upon previous work with CWYWP, this study investigated whether the incorporation of a multi-enzyme complex during fermentation could improve rumen fermentation characteristics, enhance nutrient degradability, and reduce CH_4_ production. The objective of this study was to evaluate the effects of replacing SBM with CWYWEP on in vitro rumen fermentation, gas production kinetics, and nutrient digestibility. It was hypothesized that CWYWEP would enhance fermentation efficiency, increase VFA production, improve DM degradability, and reduce CH_4_ emissions compared to SBM.

## 2. Materials and Methods

This study was conducted following the ethical guidelines for animal experimentation established by the National Research Council of Thailand. The use of animals for rumen fluid collection was approved by the Animal Ethics Committee of Khon Kaen University (approval number: IACUC-KKU-28/68). All procedures involving animals complied with institutional regulations and relevant national guidelines. The experimental animals used for rumen fluid collection were maintained under the care of the Department of Animal Science, Faculty of Agriculture, Khon Kaen University. Informed consent and institutional permission were obtained before sample collection.

### 2.1. Dietary Preparation

Yeast waste (YW), a by-product of ethanol production, was obtained from Mitr Phol Bio-Power Co., Ltd. (Chaiyaphum, Thailand), while citric waste (CW), derived from cassava wastewater during citric acid processing, was collected from Sam Mor Farm (Udon Thani, Thailand). Commercial-grade molasses (M Molasses; Do Home Co., Ltd., Khon Kaen, Thailand), urea, and citric acid powder were sourced from local suppliers. A multi-enzyme complex (Agal-Pro; New Normal Vet, Thailand) was used to enhance fiber degradation during fermentation. The enzyme blend contained alpha-galactosidase (8000 U/g), xylanase (300,000 U/g), beta-mannanase (60,000 U/g), beta-glucanase (44,000 U/g), cellulase (35,000 U/g), and pectinase (10,000 U/g).

To prepare the fermentation medium, 100 mL of YW was combined with a solution consisting of 20 g of molasses and 50 g of urea dissolved in 100 mL of distilled water, yielding a 1:1 mixture. This solution was aerated for 16 h at ambient temperature using an electromagnetic air compressor. The pH was adjusted to 3.9–4.5 using formic acid to optimize microbial growth conditions.

Two types of fermented products were subsequently formulated. The first, referred to as CWYWP, was prepared by mixing 1 kg of CW with 1 L of the YW fermentation medium. The second product, CWYWEP, was produced by first mixing 1 kg of CW with 2 g of the enzyme complex. Then, 1 L of the fermentation medium was added to the mixture.

Each mixture was packed into plastic bags, vacuum-sealed using a commercial vacuum sealer (IMAFLEX 1400 W VS-921; Imarflex Industrial Co., Ltd., Bangkok, Thailand), and incubated under anaerobic conditions at room temperature for 14 days. After fermentation, the products were sun-dried for 48 h until the moisture content was reduced to below 10%. To minimize microbial contamination, sun-drying was performed on clean concrete surfaces under mesh covers. The dried material was then pelletized using a mechanical pelletizer following a modified method adapted from Kanakai et al. [10], ensuring uniform pellet size and stability suitable for dietary inclusion.

### 2.2. Experimental Design and Feed Composition

The study employed a completely randomized design (CRD) with four replicates per treatment. Seven dietary treatments were formulated to evaluate the effects of replacing SBM with citric waste-based protein sources on rumen fermentation. Two types of pellets were used: CWYWP, produced by fermenting citric waste with yeast waste, and CWYWEP, which was further supplemented with a multi-enzyme complex. T1 served as the control diet containing 100% SBM. T2, T3, and T4 replaced SBM with CWYWP at 33%, 66%, and 100% of DM, respectively. Similarly, T5, T6, and T7 replaced SBM with CWYWEP at the corresponding levels of 33%, 66%, and 100% of DM. All diets were formulated to be isonitrogenous and isoenergetic to ensure that any observed effects could be attributed to the type and level of protein source substitution.

All feed ingredients were dried at 60 °C and ground to pass through a 1 mm screen using a Cyclotech Mill (Tecator, Sweden). The ground samples were preserved for subsequent chemical analysis and in vitro gas production assays. Chemical composition analyses included measurements of DM (method ID 967.03), ash (ID 492.05), ether extract (EE; ID 445.08), and CP (ID984.13) following the procedures of the Association of Official Analytical Chemists [11]. Organic matter (OM) was calculated by subtracting ash from 100% DM. Fiber fractions, including NDF and acid detergent fiber (ADF), were analyzed using the method of Van Soest et al. [12]. The chemical composition of the experimental diets is presented in Table 1.

### 2.3. Ruminal Fluid Donors and Substrates of Inoculum

Four Thai native beef cattle bulls, aged three years and averaging 350 ± 15.0 kg body weight (BW), were used as rumen fluid donors. The animals were housed individually with free access to clean water and were adapted to a standard feeding regimen for seven days prior to collection. They received a concentrate diet at 2% of BW, divided into two daily feedings (07:00 and 16:00 h), formulated to contain 14.0% CP on a DM basis. The concentrate consisted of cassava chips, rice bran, soybean meal, palm kernel meal, corn, urea, salt, molasses, and a vitamin–mineral premix, while rice straw was offered ad libitum.

On the day of sampling, prior to the morning feeding, approximately 1500 mL of rumen fluid was collected from each animal using an oral suction technique with a stomach tube connected to a vacuum pump. The tube was inserted into the mid-rumen region, and the first portion of fluid was discarded to avoid saliva contamination. The initial pH of the buffered rumen fluid inoculum ranged from 6.80 to 7.00. Rumen fluid from the four animals was then pooled in equal volumes into pre-warmed Erlenmeyer flasks maintained at 39 °C, yielding approximately 6.0 L of mixed inoculum. The pooled fluid was immediately filtered through four layers of cheesecloth and transported to the laboratory within 15 min in insulated containers kept at 39 °C for use in the in vitro fermentation assay.

Artificial saliva was prepared according to the protocol of Menke et al. [13] and mixed with rumen fluid in a 2:1 ratio to prepare the rumen inoculum. Serum bottles containing the test substrates were pre-incubated in a water bath at 39 °C for one hour before inoculation. The 2:1 buffer-to-rumen ratio, as recommended by Menke et al. [13], was used to maintain buffering capacity and stabilize pH during incubation. Each fermentation unit consisted of 0.5 g of DM substrate incubated with 40 mL of buffered rumen fluid in a 50 mL calibrated glass bottle sealed with a rubber stopper and aluminum cap. Before inoculation, each bottle was flushed with CO_2_ and allocated into three experimental sets.

The first set consisted of 32 bottles [(4 replicates × 7 treatments) + 4 blanks] and was used to evaluate gas production kinetics and cumulative gas volume over a 96-h incubation period. The second set included 48 bottles [(3 replicates × 7 treatments × 2 sampling times at 2 and 4 h) + 3 blanks] for measuring rumen fermentation characteristics, including pH, ammonia nitrogen (NH_3_–N), VFAs, and protozoal counts. Samples were collected at 2 and 4 h post-inoculation to capture the early fermentation phase, when VFA concentrations rapidly increase due to microbial metabolism of fermentable substrates. These time points are consistent with in vivo observations of postprandial fermentation dynamics and have been widely adopted in prior studies evaluating rumen fermentation kinetics. The third set contained 48 ANKOM F57 filter bags [(3 replicates × 7 treatments × 2 sampling times at 24 and 48 h) + 3 blanks], used for nutrient digestibility assessment using the ANKOM Daisy II Incubator (ANKOM Technology Corporation, Macedon, NY, USA). Approximately 0.50 g of each feed sample was weighed into the filter bags in duplicate.

### 2.4. Sample Collection and Analysis

#### 2.4.1. Gas Production Kinetics

Cumulative gas production was recorded at 0, 1, 2, 4, 6, 12, 24, 48, 72, and 96 h of incubation. Gas volume was measured manually by reading the calibrated plunger position of each syringe at designated time points. The gas volumes from these blank bottles were subtracted from each treatment to account for background gas production before calculating cumulative values. Gas production data were fitted to a non-linear model proposed by Schofield [14], described by the following equation:Gas production = b × [1 − exp(−c × (t − L))](1)
where b represents the asymptotic gas volume (mL/g DM) corresponding to the extent of substrate degradation, c is the rate constant of gas production (h^−1^), t is incubation time (h), and L is the lag time before fermentation begins (h).

#### 2.4.2. In Vitro Degradability (IVDMD and IVOMD)

At 24 and 48 h post-inoculation, in vitro DM degradability (IVDMD) and in vitro organic matter degradability (IVOMD) were determined following the procedure of Tilley and Terry [15]. The IVDMD (%) was calculated using the following equation:IVDMD (%) = [(DM of substrate − DM of residue)/DM of substrate] × 100(2)

Organic matter digestibility was determined by igniting dried residues and calculating the difference between dried and ashed weights. IVOMD was calculated asIVOMD (%) = [(Sample OM) − (Residual OM − OM in blank residue)]/Sample OM × 100(3)

#### 2.4.3. Fermentation Characteristics

At 2 and 4 h after incubation, pH was measured immediately using a portable pH meter (HI 8424; Hanna Instruments, Singapore). For NH_3_–N and VFAs, rumen inoculum was sampled at 2 and 4 h, filtered through four layers of cheesecloth, and separated into subsamples. To analyze NH_3_–N, 18 mL of filtered rumen fluid was preserved with 2 mL of 1 M H_2_SO_4_ and centrifuged at 3000× *g* for 10 min to stop microbial activity. The supernatant was analyzed using a UV–VIS spectrophotometer (PG Instruments, London, UK).

Volatile fatty acid concentrations including acetic acid (C2), propionic acid (C3), and butyric acid (C4), were analyzed using gas chromatography (Nexis GC-2030, Shimadzu Co., Kyoto, Japan) equipped with a flame ionization detector (FID) and a capillary column (DB-WAX, 30 m × 0.25 mm × 0.25 μm; Agilent Technologies, Santa Clara, CA, USA). The injector and detector temperatures were set at 250 °C and 270 °C, respectively. The oven temperature was programmed to start at 100 °C, held for 1 min, then increased to 180 °C at a rate of 10 °C/min, and held for 3 min. Nitrogen was used as the carrier gas at a flow rate of 1.5 mL/min. Samples were injected with a volume of 1 μL using a split ratio of 10:1. Prior to injection, rumen fluid was centrifuged, and the supernatant filtered through a 0.22 μm syringe filter. VFAs were identified and quantified by comparing retention times and peak areas with those of known standards.

Methane (CH_4_) production (mmol/g DM) was estimated based on VFA concentrations at 2 and 4 h post-inoculation using the stoichiometric equation of Moss et al. [16]CH_4_ = (0.45 × C2) − (0.275 × C3) + (0.40 × C4)(4)

These time points were selected to capture early-stage fermentation dynamics and hydrogen partitioning patterns. Although this method estimates CH_4_ production theoretically, it was not compared to total gas volume. Therefore, CH_4_ values reflect relative fermentation patterns rather than actual gas output.

#### 2.4.4. Microbial Enumeration

Protozoa populations were stained using a methyl green–formalin–saline solution and counted under a microscope using a hemacytometer (depth: 0.1 mm; chamber area: 0.0025 mm^2^; ISO LAB Laborgeräte GmbH, Eschau, Germany). Total protozoal populations were quantified based on standard counting procedures.

### 2.5. Statistical Analysis

All data were analyzed using analysis of variance (ANOVA) under a completely randomized design (CRD) with three replicates per treatment. Statistical analysis was performed using the General Linear Model (GLM) procedure in SAS software version 6 (SAS Institute Inc., Cary, NC, USA) [17]. Treatment means were compared using Tukey’s Honestly Significant Difference (HSD) test, with statistical significance declared at *p* < 0.05.

To evaluate the effects of graded replacement levels of soybean meal (SBM), orthogonal polynomial contrasts were applied to test for linear (L), quadratic (Q), and cubic (C) trends within each alternative protein source (CWYWP and CWYWEP). Additionally, orthogonal contrasts were used to compare the main effects of the protein sources (control, CWYWP, and CWYWEP), enabling targeted comparisons between SBM and the two citric waste-based protein sources.

The statistical model used was as follows:Y_ij_ = μ + α_i_ + τ_i_ + ε_ij_(5)
where Y_ij_ is the observed value from replicate j under treatment i; μ is the overall mean; α_i_ denotes the effect of SBM replacement level (0%, 33%, 66%, 100%); τ_i_ represents the effect of the protein source (control, CWYWP, or CWYWEP); and ε_ij_ is the residual error term.

## 3. Results

### 3.1. Nutritional Composition of Diet

The chemical composition of SBM, CWYWP, and CWYWEP is presented in Table 1. All protein sources had comparable DM content, ranging from 90.32% to 91.73%, indicating uniform moisture levels following drying. Crude protein content was notably higher in the fermented products compared to SBM. CWYWEP had the highest value at 50.41%, followed closely by CWYWP at 50.06%, whereas SBM contained 44.48%. These results confirm that yeast waste fermentation, particularly when supplemented with enzymes, significantly improves protein enrichment in citric waste. In terms of fiber content, CWYWP exhibited the highest levels of NDF and ADF at 22.60% and 16.88%, respectively. Enzyme supplementation in CWYWEP effectively reduced these fractions to 19.90% NDF and 15.30% ADF, indicating improved fiber degradation. As expected, SBM had the lowest fiber concentrations (16.70% NDF and 12.55% ADF). Overall, the results demonstrate that combining yeast waste and enzyme treatment can improve both the protein concentration and fiber digestibility of citric waste, supporting its use as a viable alternative protein source in ruminant diets.

### 3.2. Gas Production Kinetics and Cumulative Gas Output

The effects of replacing SBM with CWYWP and CWYWEP on in vitro gas production kinetics are shown in Table 2 and Figure 1. Cumulative gas production at 96 h differed significantly among treatments (*p* < 0.001). The 100% CWYWEP treatment yielded the highest cumulative gas volume (93.7 mL/0.5 g DM), significantly exceeding the SBM control and all CWYWP treatments. CWYWEP at 33% and 66% replacement levels produced more gas than equivalent CWYWP diets, indicating improved fermentability from enzyme supplementation. In contrast, the 33% CWYWP inclusion resulted in the lowest gas volume (52.8 mL), suggesting limited microbial degradation in the absence of exogenous enzymes. The potential gas production (b) was also significantly affected (*p* < 0.01), with the highest value observed in the SBM control (86.44 mL/g DM), while CWYWEP at 100% and 66% inclusion maintained comparably high b-values. The lowest b-value was recorded in the 33% CWYWP group (49.90 mL/g DM), consistent with its poor fermentability. Lag time (L) also varied significantly across treatments (*p* < 0.05). The CWYWEP 66:33 treatment exhibited the shortest lag time (0.17 h), indicating rapid microbial colonization and early onset of fermentation. The 100% CWYWEP diet had the longest lag time (1.00 h), possibly due to a temporary delay in microbial adaptation at full SBM replacement.

### 3.3. In Vitro Degradability

The effects of replacing SBM with CWYWP and CWYWEP on IVDMD and IVOMD at 24 and 48 h are summarized in Table 3. IVOMD values were calculated after correcting for ash content, explaining the difference with IVDMD. A significant treatment effect was observed for IVDMD at 48 h (*p* < 0.01). The CWYWEP 0:100 group had the highest IVDMD (64.49%), followed by the 66:33 and 33:66 levels (63.23%), all statistically similar to the SBM control (63.95%; *p* > 0.05). In contrast, all CWYWP groups showed significantly lower IVDMD values (ranging from 55.38% to 57.14%) compared with SBM, highlighting the limited digestibility of non-enzyme-treated citric waste. IVOMD values followed a similar trend and were significantly affected by treatment (*p* < 0.01). The CWYWEP 0:100 treatment yielded the highest IVOMD (61.56%), which did not differ significantly from the SBM control (61.73%; *p* > 0.05). In comparison, all CWYWP treatments resulted in lower IVOMD, further emphasizing the role of enzymatic supplementation in improving organic matter digestibility. Overall, these results demonstrate that CWYWEP, particularly at moderate to full replacement levels, can serve as a suitable substitute for SBM without compromising degradability. Conversely, CWYWP alone—without enzyme addition—showed inferior digestibility, reinforcing the necessity of enzyme enhancement to improve the nutritional value of citric waste-based feed ingredients.

### 3.4. Ruminal pH, Ammonia Nitrogen (NH_3_–N), and Protozoan Population

The effects of replacing SBM with CWYWP or CWYWEP on ruminal pH, NH_3_–N, and protozoal counts at 2 and 4 h of incubation are presented in Table 4. The pH of the buffered rumen fluid inoculum prior to incubation ranged from 6.80 to 7.00. After 2 and 4 h of incubation, pH values increased slightly across all treatments, ranging from 7.02 to 7.14. NH_3_–N concentrations showed a numerical increase in the CWYWEP treatments, with the highest value observed in the 0:100 group (20.68 mg/dL). However, these differences were not statistically significant (*p* > 0.05), suggesting that nitrogen release from CWYWEP was comparable to that of SBM. Protozoal populations were also unaffected by the dietary treatments (*p* > 0.05), indicating that partial or full replacement of SBM with fermented citric waste—regardless of enzyme supplementation—did not negatively impact rumen protozoal activity or microbial balance.

### 3.5. Ruminal Volatile Fatty Acid (VFA) Profile

The effects of replacing SBM with CWYWP and CWYWEP on ruminal VFA profiles and estimated CH_4_ production are summarized in Table 5. At 4 h, the CWYWEP 0:100 treatment had the highest TVFA concentration (99.63 mM/L), suggesting improved fermentation and microbial activity with full SBM replacement. In contrast, TVFA concentrations in all CWYWP treatments did not differ significantly from the control, suggesting limited enhancement of fermentation without enzyme supplementation.

Methane production estimated from VFA proportions at 2 h was significantly reduced in all CWYWEP treatments (*p* < 0.05), with the 0:100 group showing the lowest value (17.90 mmol/mol), corresponding to a 5% decrease compared to the SBM control (18.85 mmol/mol). No significant reduction in CH_4_ was observed in the CWYWP groups. This decline in CH_4_ production among CWYWEP-fed groups was associated with a favorable shift in VFA composition. Specifically, CWYWEP inclusion increased both C3 and C2 concentrations while reducing C2/C3 from 2.94 in the control to 2.66 in the 0:100 CWYWEP group. This shift indicates better hydrogen use and more efficient fermentation, likely due to yeast fermentation combined with enzymatic fiber breakdown.

## 4. Discussions

### 4.1. Nutritional Composition of Diet

Analysis showed that CWYWP and CWYWEP had higher crude protein levels than SBM—49.0% and 50.4% DM, respectively. These results are consistent with previous findings by Kanakai et al. [10], who demonstrated the protein-enriching potential of yeast-fermented agro-industrial by-products. Similarly, Suriyapha et al. [6] reported a CP concentration as high as 53.5% DM in CWYW, highlighting the role of yeast biomass accumulation in elevating protein levels, given the inherently high protein content of yeast cells.

In addition to protein synthesis, some yeast strains are known to secrete fibrolytic enzymes such as cellulase, which facilitate partial degradation of plant cell walls. This enzymatic activity not only reduces the fiber content but also concentrates the protein fraction relative to total DM [6,8]. The slightly lower CP content observed in CWYWP compared to the values reported by Suriyapha et al. [6] may reflect differences in fermentation conditions, yeast strain efficacy, or variability in substrate composition derived from ethanol production.

Notably, the inclusion of a multi-enzyme complex during CWYWEP production significantly reduced fiber fractions. CWYWEP exhibited lower NDF and ADF values (33.00% and 25.50% DM, respectively) compared to CWYWP (37.70% and 26.90% DM), indicating enhanced fiber degradation. This improvement is attributed to the synergistic action of cellulase, xylanase, and pectinase, which hydrolyze key components of the plant cell wall—including cellulose, hemicellulose, and pectins—thereby releasing soluble sugars and reducing total fiber content [18,19].

Furthermore, enzyme supplementation is known to facilitate pre-digestion of fibrous materials, increasing substrate accessibility to rumen microbes. Firkins et al. [20] reported that fibrolytic enzymes enhance microbial colonization by disrupting the structural integrity of the forage matrix. Supporting this, Song et al. [21] demonstrated that enzyme-treated tropical forages showed improved digestibility and nutrient release. These findings indicate that CWYWEP enhances citric waste’s protein content and digestibility, making it a promising protein source for ruminants.

### 4.2. Gas Production Kinetics and Cumulative Gas Output

Although increased gas production may represent energy loss, when interpreted alongside degradability and VFA profiles, it provides insight into fermentation kinetics and substrate utilization efficiency. The results indicated that replacing SBM with CWYWEP significantly improved cumulative gas production during in vitro fermentation. The highest gas volume was recorded in the 100% CWYWEP group, reaching 93.70 mL/0.5 g DM at 96 h, which was notably greater than the SBM control. In contrast, diets containing CWYWP—particularly at partial replacement levels—produced substantially lower gas volumes, underscoring its limited fermentability. These results support previous findings by Kanakai et al. [10], who reported only modest improvements in fermentation when using citric waste fermented with yeast alone. The comparison demonstrates the additional benefit of enzyme supplementation in enhancing substrate degradability and fermentative efficiency.

The increased gas output observed in CWYWEP treatments is likely due to the activity of the multi-enzyme complex applied during fermentation. Enzymes such as cellulase, xylanase, and mannanase facilitate the hydrolysis of structural carbohydrates, including cellulose and hemicellulose, into fermentable sugars, thereby improving microbial accessibility [21]. This enzymatic pre-treatment enhances substrate solubilization, which contributes to faster and more extensive microbial colonization, resulting in higher gas production.

Moreover, the increase in cumulative gas output reflects a higher degree of substrate utilization. This observation aligns with earlier studies showing that enzymatic treatment improves the digestibility of fiber fractions such as NDF and ADF, which are typically recalcitrant in untreated fibrous feeds [22,23]. Similar trends were reported by Meale et al. [18], who found that fibrolytic enzymes enhance fermentation kinetics and gas production in high-fiber diets. Beigh et al. [9] also demonstrated that enzyme-producing fungi and yeast contribute to lignocellulosic degradation, further improving nutrient availability.

### 4.3. In Vitro Degradability

Replacing SBM with CWYWEP significantly improved IVDMD and IVOMD, especially at higher inclusion levels. At 100% replacement, CWYWEP achieved the highest IVDMD (64.49%) and IVOMD (61.56%) values that were comparable to—or even slightly exceeded—those of the SBM control. In contrast, all diets containing CWYWP exhibited significantly lower degradability, underscoring the crucial role of enzyme supplementation in enhancing the nutritive quality of citric waste-based feed.

This improvement in degradability is likely due to the synergistic effects of enzymatic fiber hydrolysis and the biological activity of yeast during fermentation. The enzyme complex used in CWYWEP—comprising cellulase, xylanase, and pectinase—effectively targets structural polysaccharides such as cellulose and hemicellulose, breaking them down into smaller, more soluble substrates that are more readily fermented by rumen microbes [24].

Simultaneously, the presence of yeast contributes to improved fermentation through both nutritional enrichment and microbial stimulation. Yeast cells, particularly those in yeast waste, are rich in nitrogenous compounds, B-vitamins, and organic acids, which are known to support the growth and activity of ruminal microbes [6]. Moreover, *Saccharomyces cerevisiae*—commonly found in yeast waste—has been reported to enhance fibrolytic bacterial populations, stabilize rumen pH, and scavenge oxygen, thereby promoting an optimal anaerobic environment for fiber digestion [25]. Additionally, yeast may produce endogenous enzymes during fermentation that contribute further to partial fiber breakdown [6,24].

Fermentation may also modify the substrate’s lignocellulosic structure, increasing solubility and microbial accessibility. These combined effects—exogenous enzymatic hydrolysis, yeast-driven biochemical modification, and improved microbial adaptation—likely explain the superior IVDMD and IVOMD observed in CWYWEP diets. These findings are in agreement with those of Song et al. [21], who reported enhanced digestibility of organic matter, NDF, and ADF in rations supplemented with enzymes. Moreover, the highest IVOMD in CWYWEP treatments coincided with NH_3_–N concentrations within the optimal 15–25 mg/dL range, suggesting that ammonia release was well matched to carbohydrate degradation [20]. This synchrony would have supported efficient microbial protein synthesis, thereby amplifying OM digestibility [24,25].

In conclusion, the co-fermentation of citric waste with yeast waste and exogenous enzymes markedly improves nutrient degradability, making CWYWEP a highly digestible, protein-rich alternative to SBM in ruminant feeding systems. By contrast, CWYWP—despite its high protein content—lacks sufficient structural breakdown, limiting its effectiveness as a standalone replacement. These results highlight the importance of enzyme supplementation in maximizing the feeding value of citric waste-based protein sources.

### 4.4. Ruminal pH, NH_3_-N, and Protozoal Population

In this study, replacing SBM with CWYWP or CWYWEP did not significantly affect rumen pH, NH_3_–N concentrations, or protozoal populations at 2 and 4 h post-incubation. The observed pH values after 4 h of incubation (7.02–7.14) may result from several factors. The high crude protein content of SBM, CWYWP, and CWYWEP (44–50% CP) likely contributed to increased NH_3_–N from fermentable protein or NPN, raising pH. The buffering strength of McDougall’s solution may have also limited pH decline despite fermentation. Moreover, the short fermentation time (2–4 h) may not have allowed sufficient acid buildup. Notably, a pH slightly above 7.0 supports fibrolytic microbes such as *Ruminococcus* and *Fibrobacter*, enhancing fiber degradation in CWYWEP treatments [25].

The unchanged NH_3_–N concentrations, ranging from 16.13 to 20.68 mg/dL, indicate that protein degradation and microbial NH_3_–N utilization occurred at rates consistent with normal rumen fermentation [26]. Although NH_3_–N levels were not significantly different among treatments, the improved gas and VFA production observed in CWYWEP groups suggest better synchrony between nitrogen and energy release, potentially enhancing microbial efficiency rather than increasing degradable protein content [4,6]. According to Cherdthong et al. [26], NH_3_–N concentrations between 15 and 25 mg/dL are considered sufficient to support optimal microbial protein synthesis in the rumen. The inclusion of yeast-fermented citric waste likely provided readily available nitrogenous compounds through microbial biomass and residual urea, contributing to sustained NH_3_–N levels [6,24]. When considered alongside IVOMD results, the stable NH_3_–N concentrations in CWYWEP treatments suggest that ammonia release and OM degradation proceeded in concert. This coordination would have promoted efficient microbial protein synthesis and thereby contributed to the higher IVOMD values recorded [20]. However, to confirm changes in rumen-degradable or undegradable protein fractions, further direct measurements of RDP and RUP would be required. The absence of significant differences among treatments suggests that the nitrogen released from the CWYWEP and CWYWP was efficiently utilized by rumen microbes, maintaining a dynamic equilibrium between deamination and microbial assimilation [6,20].

Moreover, protozoal populations (ranging from 1.83 × 10^5^ to 2.93 × 10^5^ cells/mL) were not significantly influenced by treatment. This stability implies that neither the fermentation substrate nor the enzyme supplementation exerted inhibitory effects on protozoal survival or activity. Protozoa are sensitive to both low pH and abrupt changes in substrate availability, and their population dynamics often reflect broader shifts in the rumen microbial ecosystem [25]. The maintenance of protozoal abundance further supports the conclusion that CWYWEP and CWYWP inclusions were well tolerated within the rumen environment. While this study assessed protozoal populations, future work should include bacterial community profiling to better elucidate microbial dynamics.

These findings are consistent with those of Cherdthong et al. [26], who reported that substituting soybean meal (SBM) with post-fermentative yeast biomass in Thai native cattle had no adverse effects on rumen pH, NH_3_–N concentrations, or protozoal populations. This stability may be attributed to the beneficial effects of yeast-fermented feeds, which can enhance fiber digestion, support balanced nitrogen metabolism, and help maintain a stable microbial ecosystem through probiotic action [6,10]. Collectively, the evidence suggests that both CWYWP and CWYWEP are viable alternatives to SBM in ruminant diets, capable of sustaining key rumen fermentation parameters critical for microbial activity and overall rumen function.

### 4.5. Ruminal Volatile Fatty Acid (VFA) Profile

The increase in VFA concentrations between 2 and 4 h reflects additive fermentation output beyond the baseline, consistent with rapid carbohydrate fermentation. Replacing SBM with CWYWEP had a pronounced effect on rumen fermentation, enhancing total VFA production and reducing estimated CH_4_ emissions. Notably, the CWYWEP 0:100 and 33:66 treatments increased total VFA concentrations by approximately 5% compared to the SBM control, indicating improved carbohydrate fermentability. This enhancement is likely attributable to the enzymatic degradation of lignocellulosic components during fermentation, which increases carbohydrate solubility and microbial accessibility [24]. In contrast, CWYWP showed no such improvement, suggesting that yeast fermentation alone did not significantly enhance fiber digestibility or fermentation extent. Furthermore, measured protozoal counts were lower in CWYWEP treatments, complementing the shift toward C3 and suggesting protozoal-mediated hydrogen transfer to C3 pathways [25,26,27].

The inclusion of CWYWEP notably shifted the rumen fermentation profile, marked by increased C3 and decreased C2 proportions, particularly in the group receiving 100% replacement. This shift reduced the C2/C3 ratio by 10%, suggesting more efficient hydrogen use through C3 pathways that reduce methane formation [27,28]. Similar trends have been reported by Kanakai et al. [10] and Cherdthong et al. [26], who observed enhanced C3 levels following supplementation with CWYWP and yeast waste-based products. These effects are likely attributed to fermentation conditions that promote the growth of amylolytic, C3-producing bacteria. Since C3 formation competes with methanogenesis for hydrogen, the shift effectively redirects H_2_ from methane production. As Jeong et al. [29] noted, increasing C3 production can limit the hydrogen available for methane formation. This offers a practical approach to methane mitigation by shifting fermentation patterns rather than directly inhibiting methanogens [30,31]. One limitation of this study is the absence of fermentation product measurements at 0 h. This decision was based on the immediate onset of microbial activity upon inoculation and the technical difficulty of capturing a representative baseline without compromising anaerobic conditions. Future studies could consider incorporating time-zero sampling by pre-analyzing the buffered rumen fluid and modeling expected dilution effects to better characterize net metabolite changes.

Methane production exhibited a dose-dependent reduction with increasing CWYWEP inclusion. The greatest decline—over 5% compared to the SBM control—was observed in the CWYWEP 0:100 group, while intermediate reductions were found in the 33:66 and 66:33 groups. In contrast, CH_4_ output remained unchanged in all CWYWP treatments, further reinforcing the critical role of enzyme supplementation in modulating fermentation end-products. The lower protozoal abundance in CWYWEP diets further supports the redirection of hydrogen away from methanogenesis and toward C3 formation. This CH_4_ reduction is likely due to redirected hydrogen flow, enhanced carbohydrate digestibility [32], and, possibly, reduced pH. These changes contribute to reduced methanogenic activity [27,30].

The combined action of yeast and multi-enzyme supplementation during citric waste fermentation supports several synergistic mechanisms that collectively reduce CH_4_ production [33]. Enzymes such as cellulase, xylanase, and pectinase hydrolyze fibrous components—cellulose, hemicellulose, and pectin—thereby releasing fermentable sugars and shifting fermentation toward increased C3 production, which competes with methanogenesis for hydrogen [34]. These findings are consistent with Suriyapha et al. [6], who observed reduced CH_4_ output when replacing 75% of SBM with citric waste fermented with yeast waste under in vitro conditions.

While the current findings offer strong in vitro evidence of CWYWEP’s potential to enhance fermentation efficiency and mitigate CH_4_ emissions, further in vivo studies are needed to confirm its effects on animal performance, palatability, and economic viability under practical feeding conditions.

## 5. Conclusions

This study demonstrates that CWYWEP can fully replace soybean meal (SBM) in concentrate-based diets without impairing rumen function. CWYWEP enhanced in vitro gas production, dry matter and organic matter degradability, and shifted fermentation toward greater C3 production, resulting in reduced estimated CH_4_ emissions. These improvements were not observed with CWYWP, highlighting the critical role of enzyme supplementation in enhancing nutrient utilization. Ruminal pH, NH_3_–N concentrations, and protozoal populations remained stable across treatments, indicating no adverse effects on rumen microbial balance. CWYWEP appears to be a sustainable, high-protein alternative to SBM, supporting fermentation and climate-smart feeding strategies. Further in vivo studies are warranted to validate these results under practical conditions and to assess the long-term impacts on animal performance, feed economics, and environmental outcomes. Further research should establish quality control benchmarks to ensure consistency and safety in large-scale CWYWEP production.

## Figures and Tables

**Figure 1 animals-15-02351-f001:**
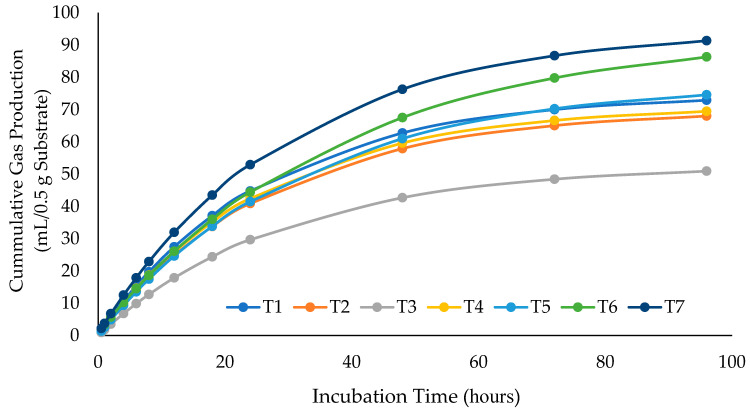
Effect of the different main protein sources on cumulative gas production after 0–96 h of incubation (T1: control group containing 100% SBM; T2: 33% of SBM was replaced with CWYWP; T3: 66% was replaced with CWYWP; T4: SBM was completely replaced by CWYWP; T5: contained 33% CWYWEP in place of SBM; T6: included 66% CWYWEP; and T7: 100% of SBM was replaced by CWYWEP).

**Table 1 animals-15-02351-t001:** Ingredient and chemical composition of the concentrate diet used in the experiment.

Item	Control	Replacement of SBM (%)						Rice Straw	CWYWP	CWYWEP
	SBM, 100%	SBM/CWYWP			SBM/CSYWEP					
		66:33	33:66	0:100	66:33	33:66	0:100			
Ingredients (% DM)										
Cassava chips	50	50	50	50	50	50	50			
Rice bran	10	10	10	10	10	10	10			
Soybean meal	15	10	5	0	10	5	0			
CWYWP	0	5	10	15	0	0	0			
CWYWEP	0	0	0	0	5	10	15			
Palm kernel meal	11	11	11	11	11	11	11			
Corn	7.7	7.8	7.9	8	8	8.1	8.2			
Urea	1.3	1.2	1.1	1	1	0.9	0.8			
Mineral premix	1	1	1	1	1	1	1			
Molasses, liquid	2.5	2.5	2.5	2.5	2.5	2.5	2.5			
Salt	1	1	1	1	1	1	1			
Pure sulfur	0.5	0.5	0.5	0.5	0.5	0.5	0.5			
Chemical composition										
Dry matter (%)	93.30	93.40	93.70	93.60	93.70	94.60	93.30	94.10	93.10	93.40
	----------------% Dry matter---------------									
Organic matter	96.50	96.30	95.80	95.40	96.10	94.40	95.10	99.30	90.10	89.30
Ash	3.50	3.70	4.20	4.60	3.90	5.60	4.90	0.70	9.90	10.70
Crude protein	14.30	14.40	14.30	14.30	14.30	14.40	14.40	2.80	49.00	50.40
Ether extract	3.90	3.80	3.60	3.60	3.30	3.60	3.40	0.60	1.80	1.90
Neutral detergent fiber	19.30	21.60	22.40	22.20	17.80	17.80	21.90	71.20	37.70	33.00
Acid detergent fiber	10.30	13.20	14.60	17.20	13.00	14.20	14.70	45.30	26.90	25.50

Premix (1 kg) contains the following vitamins and minerals: vitamin A: 10,000,000 IU; vitamin E 70,000 IU; vitamin D: 1,600,000 IU; Fe: 50 g; Zn: 40 g; Mn: 40 g; Co: 0.1 g; Se: 0.1 g; I: 0.5 g. SBM = soybean meal; CWYWP = citric waste fermented with yeast waste in pellet form; CWYWEP = citric waste fermented with yeast waste with enzymes in pellet form.

**Table 2 animals-15-02351-t002:** In vitro kinetics and accumulation of gas after 96 h of incubation time.

Protein Source	SBM Replacement Ratio	Gas Kinetics			Cumulative Gas (96 h) mL/0.5 g DM Substrate
		b	c	L	
Control	100:0	77.53 ^a^	0.055	0.28 ^c^	82.60 ^a^
CWYWP	66:33	60.98 ^bc^	0.052	0.27 ^c^	67.80 ^bc^
	33:66	41.99 ^c^	0.049	0.30 ^bc^	52.80 ^b^
	0:100	67.03 ^a^	0.049	0.33 ^b^	70.20 ^ab^
CWYWEP	66:33	62.98 ^ab^	0.046	0.17 ^d^	76.60 ^ab^
	33:66	67.49 ^a^	0.044	0.65 ^ab^	85.40 ^a^
	0:100	71.43 ^a^	0.049	1.00 ^a^	93.70 ^a^
SEM		6.28	0.010	0.14	7.79
Orthogonal polynomials					
Linear		<0.05	0.29	<0.05	0.06
Quadratic		0.11	0.70	0.46	<0.05
Cubic		0.53	0.51	0.43	0.17
Orthogonal contrast					
Control vs. CWYWP		0.74	0.96	0.52	0.73
Control vs. CWYWEP		0.37	0.80	0.91	0.61
CWYWP vs. CWYWEP		0.53	0.29	<0.05	<0.05

SBM = soybean meal; CWYWP = citric waste fermented with yeast waste in pellet form; CWYWEP = citric waste fermented with yeast waste with enzymes in pellet form. Gas production = b [1 − exp^−c (t − L)^]; t = the incubation time (hours); c = a rate constant (per time unit); L = a discontinuous lag term (h); b = the final asymptotic gas volume corresponding to fully digested substrate (mL/g DM). ^a–c^ values in the same column with different superscripts differ (*p* < 0.05); SEM = standard error of the mean.

**Table 3 animals-15-02351-t003:** Effects of replacing soybean meal (SBM) with citric waste-based protein sources (CWYWP and CWYWEP) on in vitro degradability at 24 h and 48 h of incubation.

Protein Source	SBM Replacement Ratio	IVDMD (% DM)		IVOMD (% DM)	
		24 h	48 h	24 h	48 h
Control	100:0	40.99	56.11 ^a^	78.48	86.78
CWYWP	66:33	39.52	53.97 ^b^	79.17	84.93
	33:66	40.37	54.21 ^b^	79.48	86.52
	0:100	39.64	56.21 ^ab^	78.96	86.85
CWYWEP	66:33	41.64	59.11 ^a^	78.83	87.11
	33:66	42.85	59.02 ^a^	79.54	87.10
	0:100	41.19	61.32 ^a^	79.83	88.12
SEM		1.35	1.53	0.51	1.11
Orthogonal polynomials					
Linear		0.58	0.07	0.12	0.96
Quadratic		0.78	<0.05	0.99	0.34
Cubic		0.50	0.19	0.47	0.17
Orthogonal contrast					
Control vs. CWYWP		0.46	<0.01	0.72	0.34
Control vs. CWYWEP		0.86	0.50	0.31	0.29
CWYWP vs. CWYWEP		0.07	<0.01	0.61	0.90

SBM = soybean meal; CWYWP = citric waste fermented with yeast waste in pellet form; CWYWEP = citric waste fermented with yeast waste with enzymes in pellet form; IVDMD = in vitro dry matter degradability, IVOMD = in vitro organic matter degradability. ^a,b^ values in the same column with different superscripts differ (*p* < 0.05); SEM = standard error of the mean.

**Table 4 animals-15-02351-t004:** Effects of replacing soybean meal (SBM) with citric waste-based protein sources (CWYWP and CWYWEP) on ruminal pH, ammonia nitrogen (NH_3_–N), and protozoal populations at 2 and 4 h of incubation.

Protein Source	SBM Replacement Ratio	pH		Ammonia Nitrogen Concentration (mg/dL)		Protozoal Count (×10^5^ cell/mL)	
		2 h	4 h	2 h	4 h	2 h	4 h
Control	100:0	7.05	7.02	17.13	18.99	2.33	2.33
CWYWP	66:33	7.05	7.07	16.13	17.27	2.67	2.80
	33:66	7.12	7.03	16.63	17.18	2.67	2.43
	0:100	7.14	7.03	17.03	17.34	2.33	2.80
CWYWEP	66:33	7.07	7.02	16.49	18.64	2.33	2.93
	33:66	7.05	7.04	16.58	18.58	2.00	2.83
	0:100	7.06	7.03	17.09	19.06	1.83	2.60
SEM		0.02	0.02	0.49	0.98	0.37	0.28
Orthogonal polynomials							
Linear		0.74	0.83	0.79	0.43	0.25	0.37
Quadratic		0.10	0.65	0.24	0.15	0.91	0.24
Cubic		0.41	0.23	0.14	0.37	0.58	0.68
Orthogonal contrasts							
Control vs. CWYWP		0.16	0.81	0.71	0.27	0.36	0.18
Control vs. CWYWEP		0.68	0.65	0.90	0.96	0.24	0.43
CWYWP vs. CWYWEP		0.06	0.57	0.74	0.08	0.09	0.64

SBM = soybean meal; CWYWP = citric waste fermented with yeast waste in pellet form; CWYWEP = citric waste fermented with yeast waste with enzymes in pellet form; SEM = standard error of the mean.

**Table 5 animals-15-02351-t005:** Effects of replacing soybean meal (SBM) with citric waste-based protein sources (CWYWP and CWYWEP) on ruminal volatile fatty acid (VFA) profiles and methane production at 2 h and 4 h of incubation.

Protein Source	SBM Replacement Ratio	Total VFA (mM/L)		Acetic Acid (C2% of TVFA)		Propionic Acid (C3% of TVFA)		Butyric Acid (C4% of TVFA)		C2/C3 Ratio		Methane (mmol/g DM)	
		2 h	4 h	2 h	4 h	2 h	4 h	2 h	4 h	2 h	4 h	2 h	4 h
Control	100:0	76.98	81.55 ^ab^	64.67	65.73 ^cd^	22.34 ^a^	21.93	12.99	12.34	2.90	2.99	27.7 ^ab^	28.03
CWYWP	66:33	76.48	79.17 ^b^	65.69	64.94 ^d^	21.85 ^b^	22.56	12.46	12.50	3.01	3.00	28.0 ^b^	27.58
	33:66	75.79	79.11 ^b^	66.06	66.05 ^cd^	21.58 ^b^	21.82	12.36	12.13	3.06	3.03	28.2 ^b^	28.01
	0:100	78.65	86.42 ^ab^	64.98	67.11 ^ab^	21.92 ^ab^	21.42	13.10	11.47	2.99	3.13	28.0 ^b^	28.41
CWYWEP	66:33	84.44	98.57 ^a^	65.44	67.62 ^a^	22.27 ^a^	21.16	12.29	11.22	2.92	3.19	27.7 ^ab^	28.61
	33:66	81.35	98.13 ^a^	65.10	66.72 ^abc^	22.39 ^a^	21.80	12.51	11.48	2.81	3.06	27.7 ^ab^	28.15
	0:100	80.71	99.63 ^a^	64.57	65.57 ^cd^	23.03 ^a^	22.45	12.40	11.98	2.90	2.92	27.4 ^a^	27.69
SEM		2.61	5.96	0.49	0.36	0.30	0.22	0.56	1.00	0.06	0.05	1.30	0.03
Orthogonal polynomials													
Linear		0.78	0.06	0.40	<0.05	0.07	0.59	0.72	0.07	0.11	0.23	0.15	0.35
Quadratic		0.13	0.14	0.17	<0.05	0.26	0.05	0.20	0.31	0.20	0.05	0.21	0.09
Cubic		0.14	0.32	0.57	0.07	0.84	0.09	0.73	0.20	0.88	0.07	0.73	0.05
Orthogonal contrasts													
Control vs. CWYWP		0.31	<0.05	0.46	0.17	0.47	0.70	0.21	0.10	0.48	0.44	0.58	0.49
Control vs. CWYWEP		0.89	0.22	0.81	0.61	0.87	0.51	0.81	0.13	0.66	0.80	0.52	0.98
CWYWP vs. CWYWEP		0.13	<0.05	0.06	0.06	<0.05	0.48	0.19	0.06	0.05	0.23	<0.05	0.22

SBM = soybean meal; CWYWP = citric waste fermented with yeast waste in pellet form; CWYWEP = citric waste fermented with yeast waste with enzymes in pellet form; ^a–d^ values in the same column with different superscripts differ (*p* < 0.05); SEM = standard error of the mean.

## Data Availability

The original data presented in this study are included in the article and. Further inquiries can be directed to the corresponding author(s).

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
