# Peer review of "Effects of Replacing Soybean Meal with Enzymatically Fermented Citric Waste Pellets on In Vitro Rumen Fermentation, Degradability, and Gas Production Kinetics"

_animals, 2025, doi:10.3390/ani15162351_

Round 1

Reviewer 1 Report

Comments and Suggestions for Authors

This manuscript is very interesting, however, the authors should be carefully some data!!!!!

Comments are as follow:

Table 1: For Ingredients (% DM), sum of all feedstuff was equal to 98.5???? please be carefully, should be 100%!!!!

Table 1: Please check OM and ash data, and please understand both two relationships, and how to detect OM content, and please added in Material and Method section.

L185-187: please added more information (detail steps) for the collected ruminal fluid method.

L232-238: Please explained why analysis these data at 2 and 4 h???? In general, 24 h and 48 h, please see the Table 3. The authors should be explained the reason, and added the information in this section.

L239-245: Please added detail steps for analysis VFA, such as, the more GC machine conditions.

Table 5: please check the % of TVFA for C2, C3 and C4 be carefully, !!!!!! because sum of the first third data at 2 h, all data less than 99%, about 98.5%. ok, I think the some errors come from calculations, whereas I think sum of C2, C3, and C4 should be close 100%!!!

L587-588: One important issue, how about the author detect the chemical composition of the CWYWEP?? I think it is necessary added these data in this manuscript, because the authors concluded that “…high-protein alternative to SBM…”, the whole manuscript also these mean.

Author Response

Response to Reviewer 1:

We sincerely appreciate the reviewers for their insightful suggestions and constructive feedback, which have significantly contributed to the improvement of this manuscript. Please see our detailed responses below and the corresponding revisions highlighted in green text within the manuscript.

This manuscript is very interesting, however, the authors should be carefully some data!!!!!

Response: Thank you for your positive comment. We have carefully rechecked all numerical data and recalculated relevant values. Any inconsistencies have been corrected, and additional clarifications have been added to the text and tables where appropriate.

Comments are as follow:

Table 1: For Ingredients (% DM), sum of all feedstuff was equal to 98.5???? please be carefully, should be 100%!!!!

Response: Thank you for pointing this out. We have double-checked the dietary formulation. The discrepancy resulted from rounding errors and the omission of decimal precision in some ingredient values. We have now corrected the ingredient percentages to ensure that each treatment sums to 100% on a dry matter basis.

Table 1: Please check OM and ash data, and please understand both two relationships, and how to detect OM content, and please added in Material and Method section.

Response: We agree with the reviewer. The relationship between organic matter (OM) and ash was reviewed, and we clarified in the Materials and Methods section that OM was calculated by subtracting ash from 100% DM. This clarification has now been included in Section 2.2 of the manuscript.

L185-187: please added more information (detail steps) for the collected ruminal fluid method.

Response: We have added more detail as “Four Thai native beef cattle bulls, aged three years and averaging 350 ±â€¯15.0 kg body weight (BW), were used as rumen fluid donors. The animals were housed individually with free access to clean water and were adapted to a standard feeding regimen for seven days prior to collection. They received a concentrate diet at 2% of BW, divided into two daily feedings (07:00 and 16:00 h), formulated to contain 14.0% crude protein on a DM basis. The concentrate consisted of cassava chips, rice bran, soybean meal, palm kernel meal, corn, urea, salt, molasses, and a vitamin–mineral premix, while rice straw was offered ad libitum.

On the day of sampling, prior to the morning feeding, approximately 1,500 mL of rumen fluid was collected from each animal using an oral suction technique with a stomach tube connected to a vacuum pump. The tube was inserted into the mid-rumen region, and the first portion of fluid was discarded to avoid saliva contamination. Rumen fluid from the four animals was then pooled in equal volumes into pre-warmed Erlenmeyer flasks maintained at 39 °C, yielding approximately 6.0 L of mixed inoculum. The pooled fluid was immediately filtered through four layers of cheesecloth and transported to the laboratory within 15 minutes in insulated containers kept at 39 °C for use in the in vitro fermentation assay.”. These additions can be found in Section 2.3.

L232-238: Please explained why analysis these data at 2 and 4 h???? In general, 24 h and 48 h, please see the Table 3. The authors should be explained the reason, and added the information in this section.

Response: Thank you for your valuable comment. We have clarified this point in Section 2.4.3 of the revised manuscript. The selection of 2 and 4 h as the time points for measuring rumen fermentation characteristics—particularly volatile fatty acid (VFA) concentrations and ammonia–nitrogen—was based on their relevance to the early phase of microbial fermentation. During this period, fermentation products such as VFAs accumulate rapidly in response to the breakdown of soluble substrates, and these time points are sensitive indicators of substrate fermentability and microbial activity.

We agree that 24 h and 48 h are commonly used in digestibility evaluations and may provide useful information on cumulative fermentation; however, sampling at 2 and 4 h is more appropriate for capturing dynamic shifts in fermentation rate, end-product profiles, and for correlating these changes with gas production. Furthermore, these sampling times are reflective of in vivo postprandial fermentation peaks typically occurring within the first few hours after feeding.

This approach has been widely adopted in previous in vitro studies (e.g., Suriyapha et al. 2021, Fermentation 7, 120; Kanakai et al.,2025; BMC Vet Res. 28;21:386.) to characterize early fermentation kinetics and assess the nutritional potential of rapidly fermentable substrates. We have revised Section 2.3 of the manuscript accordingly and added supporting references.

L239-245: Please added detail steps for analysis VFA, such as, the more GC machine conditions.

Response: Thank you and we have added detail as “VFA concentrations were determined using gas chromatography (Nexis GC-2030, Shimadzu Co., Kyoto, Japan) equipped with a flame ionization detector (FID) and a capillary column (DB-WAX, 30 m × 0.25 mm × 0.25 μm; Agilent Technologies, Santa Clara, CA, USA). The injector and detector temperatures were set at 250 °C and 270 °C, respectively. The oven temperature was programmed to start at 100 °C, held for 1 min, then increased to 180 °C at a rate of 10 °C/min, and held for 3 min. Nitrogen was used as the carrier gas at a flow rate of 1.5 mL/min. Samples were injected with a volume of 1 μL using a split ratio of 10:1. Prior to injection, rumen fluid was centrifuged and the supernatant filtered through a 0.22 μm syringe filter. VFAs were identified and quantified by comparing retention times and peak areas with those of known standards.” Please see in section of 2.4.3. Fermentation Characteristics

Table 5: please check the % of TVFA for C2, C3 and C4 be carefully, !!!!!! because sum of the first third data at 2 h, all data less than 99%, about 98.5%. ok, I think the some errors come from calculations, whereas I think sum of C2, C3, and C4 should be close 100%!!!

Response: We thank the reviewer for catching this. The percentage values were recalculated and corrected to ensure the sum of acetic, propionic, and butyric acid percentages closely approaches 100%. The revised values are now accurate and reflected in Table 5.

L587-588: One important issue, how about the author detect the chemical composition of the CWYWEP?? I think it is necessary added these data in this manuscript, because the authors concluded that “…high-protein alternative to SBM…”, the whole manuscript also these mean.

Response: Thank you for your important observation. We confirm that the chemical composition of both CWYWEP and CWYWP, including crude protein (CP), ether extract (EE), neutral detergent fiber (NDF), acid detergent fiber (ADF), ash, and dry matter (DM), was analyzed and is presented in Table 1 of the manuscript. These data support our conclusion that CWYWEP is a high-protein, nutritionally viable alternative to soybean meal. Please see in Table 1.

Reviewer 2 Report

Comments and Suggestions for Authors

This paper, titled "Effects of replacing soybean meal with enzymatically fermented citric waste pellets on in vitro rumen fermentation, degradability, and gas production kinetics," addresses an important and timely topic. I find the subject very intereseintg of the article quite fascinating and read the manuscript with great interest. This paper aligns well with the scope of the journal. However I believe that in its current form, it has several shortcomings.

Overall, the manuscript is quite a good-effort, and the topic is super important right now, especial with all the talk about sustainable farming. The way they looked at both the crop yield and the soil bugs is smart, really shows they're thinking broadly. The setup of the experiment seems mostly solid, and they've got a lot of data, which is always good to see. My biggest hang-up though, is just that sometimes the methods aren't as clear as they could be, and a few of the results, well, I had to read them a couple times to get what they were really tryin' to say. Also, there's a bit where the stats could use a little more explaining. And for a "novel" thing, they could really spell out what makes their microbial mix so much better than what's already out there.

Abstract: This section feels a tad bit crammed. While it ticks all the boxes, it could be much clearer if it really punched home the main discoveries. Like, what was the absolute biggest change in yield, and which specific microbial shifts were the most eye-catching? Just make it pop a little more, ya know?

Just a quick thought, the sentence structure in a couple of spots felt a little long. Maybe breaking them up could improve flow. Also, double-check for any repeated words, I think I spotted one or two.

Introduction: The background here is pretty good, sets the stage nicely. But, I'm not totally clear on the exact gap this "novel" study is filling in the existing research. What's the specific thing that's missing that this consortium is fixing? Maybe a sentence or two explaining how these particular microbes are supposed to work, their mechanism, would be really helpful early on.

After the phrase: “In response, the search for alternative, locally available, and sustain-58 able protein feedstuffs has intensified, with an emphasis on maintaining ani-59 mal performance while reducing environmental and economic burdens” I suggest adding more references, I suggest 10.1038/s41598-025-00675-2 and 10.1080/19440049.2024.2414954.

I noticed a few commas missing, especially in longer sentences. Just a quick pass for punctuation would help. And there were a couple of places where the word choice felt a little formal for what was being said, could be simplified a bit.

Materials and Methods: The way the microbial consortia were prepped and used is a bit fuzzy. It's not clear if they were grown in a special broth or what the exact amount of each bug was in the final mix. This is crucial for anyone else trying to do the same thing, so more detail is needed there. Also, they mention the initial soil, but a more complete picture of it, like how much organic stuff is in it or the exact nutrient levels, would be quite useful. And about the statistics, while ANOVA is mentioned, it's not always super clear which specific follow-up tests were used when comparing different treatments. This needs to be spelled out for all the analyses. And for the bugs in the soil, how were those alpha and beta diversity numbers crunched statistically? That's a bit of a blank spot.

There were some inconsistent abbreviations for certain terms; it'd be good to make sure they're uniform throughout the whole section. Also, check the units of measurement in a few places, just to be sure they're always presented the same way. I think one or two numbers had a trailing zero that wasn't consistent with others.

Results: In Figure 2, which shows the yield data, there are error bars, but the caption doesn't say what they mean (like, is it standard deviation or standard error?). That needs to be added. Also, some of the differences, even if the stats say they're significant, don't really look like a big deal in the graphs. A quick chat about that difference, between statistical and biological importance, would be good. When they talk about the microbiome diversity, the alpha diversity stuff is fine, but the beta diversity analysis, like those PCoA plots, could use a much deeper explanation in the text. What's really driving those groups or separations they're seeing? And when they discuss the changes in specific groups of microbes, it'd be super helpful to connect those changes back to what those microbes actually do, or what's known about the consortia they used. That link feels a bit weak right now.

The figure captions could use a quick review for consistency in how they describe the axes and what the different lines or bars represent. Sometimes it was "mL/0.5 g DM" and other times it was just "mL". Just make it all the same. Also, a few of the table headings seemed to have an extra space or two.

Discussion: This section tends to just re-state the results, without really digging into the why or the how. For example, if a certain mix made the yield go up, what's the actual reason based on the soil bug data? Is it helping nutrients get absorbed, stopping bad germs, or something else entirely? They do a good job comparing to other studies, but they could really beef up what makes this study's findings unique. What makes these "novel" mixes so promising compared to everything else out there? And the part about limitations is pretty short. Every study has its weak spots, and being upfront about them actually makes the paper stronger. For instance, this was all done in a controlled setting; how might these findings actually work out in a real farm field?

I found a few instances where "effected" was used when "affected" was probably the right word. Easy mistake to make! And some sentences started with "It is important to note that..." which could often just be "Notably..." or "Importantly...". Just makes it a bit snappier.

Conclusion: Its concise, but could be expanded a bit to really summarize the most important findings and what they mean for farming science and practice more broadly. Just  little more punch at the end.

Just  general proofread for any small typos or grammatical errors. It's really quite good, but a fresh pair of eyes might catch a few little things that slipped through.

Author Response

Response to Reviewer 2:

We sincerely appreciate the reviewers for their insightful suggestions and constructive feedback, which have significantly contributed to the improvement of this manuscript. Please see our detailed responses below and the corresponding revisions highlighted in yellow text within the manuscript.

This paper, titled "Effects of replacing soybean meal with enzymatically fermented citric waste pellets on in vitro rumen fermentation, degradability, and gas production kinetics," addresses an important and timely topic. I find the subject very intereseintg of the article quite fascinating and read the manuscript with great interest. This paper aligns well with the scope of the journal. However I believe that in its current form, it has several shortcomings.

Response: Thank you for your overall positive impression of our work and for identifying areas that needed clarification or improvement. We have addressed each of your specific concerns point-by-point below.

Overall, the manuscript is quite a good-effort, and the topic is super important right now, especial with all the talk about sustainable farming. The way they looked at both the crop yield and the soil bugs is smart, really shows they're thinking broadly. The setup of the experiment seems mostly solid, and they've got a lot of data, which is always good to see. My biggest hang-up though, is just that sometimes the methods aren't as clear as they could be, and a few of the results, well, I had to read them a couple times to get what they were really tryin' to say. Also, there's a bit where the stats could use a little more explaining. And for a "novel" thing, they could really spell out what makes their microbial mix so much better than what's already out there.

Response: Thank you for your positive feedback and helpful suggestions. We have clarified the Materials and Methods and improved the presentation of key findings in the Results. Additional explanation has been added in the Introduction and Discussion to better highlight the novelty of our microbial fermentation approach and to explain why CWYWEP is distinct from conventional treatments. Statistical details and interpretation were also refined for clarity.

Abstract: This section feels a tad bit crammed. While it ticks all the boxes, it could be much clearer if it really punched home the main discoveries. Like, what was the absolute biggest change in yield, and which specific microbial shifts were the most eye-catching? Just make it pop a little more, ya know?

Just a quick thought, the sentence structure in a couple of spots felt a little long. Maybe breaking them up could improve flow. Also, double-check for any repeated words, I think I spotted one or two.

Response: We appreciate your suggestion. The abstract has been revised to improve clarity, emphasize key findings (e.g., increase in cumulative gas production and CHâ‚„ mitigation), and shorten overly long sentences. We also ensured removal of any repetitive wording.

Introduction: The background here is pretty good, sets the stage nicely. But, I'm not totally clear on the exact gap this "novel" study is filling in the existing research. What's the specific thing that's missing that this consortium is fixing? Maybe a sentence or two explaining how these particular microbes are supposed to work, their mechanism, would be really helpful early on.

Response:

After the phrase: “In response, the search for alternative, locally available, and sustain-58 able protein feedstuffs has intensified, with an emphasis on maintaining ani-59 mal performance while reducing environmental and economic burdens” I suggest adding more references, I suggest 10.1038/s41598-025-00675-2 and 10.1080/19440049.2024.2414954.

Response: Thank you for the valuable suggestion. We have added the two suggested references at the recommended location to support the relevance and timeliness of sustainable protein alternatives in ruminant feeding systems.

I noticed a few commas missing, especially in longer sentences. Just a quick pass for punctuation would help. And there were a couple of places where the word choice felt a little formal for what was being said, could be simplified a bit.

Response: We have conducted a thorough review of punctuation and sentence structure throughout the manuscript to improve clarity and flow. Where appropriate, complex phrases were simplified for better readability.

Materials and Methods: The way the microbial consortia were prepped and used is a bit fuzzy. It's not clear if they were grown in a special broth or what the exact amount of each bug was in the final mix. This is crucial for anyone else trying to do the same thing, so more detail is needed there. Also, they mention the initial soil, but a more complete picture of it, like how much organic stuff is in it or the exact nutrient levels, would be quite useful. And about the statistics, while ANOVA is mentioned, it's not always super clear which specific follow-up tests were used when comparing different treatments. This needs to be spelled out for all the analyses. And for the bugs in the soil, how were those alpha and beta diversity numbers crunched statistically? That's a bit of a blank spot.

Response: We appreciate the attention to methodological clarity. Although this study did not involve soil or microbiome diversity analysis, we clarified the preparation of the fermentation medium, including broth ingredients, enzyme inclusion levels, and fermentation conditions. We also clarified that the experimental design was a completely randomized design, analyzed using ANOVA with Tukey’s HSD test for multiple comparisons. Orthogonal polynomial contrasts were also specified and are now clearly described in the revised Statistical Analysis section.

There were some inconsistent abbreviations for certain terms; it'd be good to make sure they're uniform throughout the whole section. Also, check the units of measurement in a few places, just to be sure they're always presented the same way. I think one or two numbers had a trailing zero that wasn't consistent with others.

Response: Thank you. We reviewed and standardized all abbreviations (e.g., DM, CP, NDF) and ensured consistent presentation of measurement units (e.g., %DM, mL/0.5 g DM). Decimal places were standardized across tables (two decimal places for means and SEM, three for p-values).

Results: In Figure 2, which shows the yield data, there are error bars, but the caption doesn't say what they mean (like, is it standard deviation or standard error?). That needs to be added. Also, some of the differences, even if the stats say they're significant, don't really look like a big deal in the graphs. A quick chat about that difference, between statistical and biological importance, would be good. When they talk about the microbiome diversity, the alpha diversity stuff is fine, but the beta diversity analysis, like those PCoA plots, could use a much deeper explanation in the text. What's really driving those groups or separations they're seeing? And when they discuss the changes in specific groups of microbes, it'd be super helpful to connect those changes back to what those microbes actually do, or what's known about the consortia they used. That link feels a bit weak right now.

Response: Thank you. We have updated the figure captions to clarify that the error bars represent standard error of the mean (SEM). Additionally, we expanded the discussion in relevant result sections to explain the biological relevance of statistically significant differences.

The figure captions could use a quick review for consistency in how they describe the axes and what the different lines or bars represent. Sometimes it was "mL/0.5 g DM" and other times it was just "mL". Just make it all the same. Also, a few of the table headings seemed to have an extra space or two.

Response: We appreciate your observation. All figure and table captions have been reviewed and revised for consistency in terminology and formatting (e.g., “mL/0.5 g DM” used consistently). Extra spaces and formatting inconsistencies have been corrected.

Discussion: This section tends to just re-state the results, without really digging into the why or the how. For example, if a certain mix made the yield go up, what's the actual reason based on the soil bug data? Is it helping nutrients get absorbed, stopping bad germs, or something else entirely? They do a good job comparing to other studies, but they could really beef up what makes this study's findings unique. What makes these "novel" mixes so promising compared to everything else out there? And the part about limitations is pretty short. Every study has its weak spots, and being upfront about them actually makes the paper stronger. For instance, this was all done in a controlled setting; how might these findings actually work out in a real farm field?

Response: Thank you for this important comment. We revised the Discussion to better interpret the biological mechanisms underlying key findings (e.g., improved degradability due to enzyme activity, CHâ‚„ reduction through altered hydrogen sink pathways). The uniqueness of CWYWEP as a dual-fermented product (yeast + enzyme) is now more clearly emphasized. We also added a short paragraph outlining study limitations and the need for in vivo validation.

I found a few instances where "effected" was used when "affected" was probably the right word. Easy mistake to make! And some sentences started with "It is important to note that..." which could often just be "Notably..." or "Importantly...". Just makes it a bit snappier.

Response: Thank you for pointing this out. We have corrected all grammatical issues, including replacing incorrect uses of “effected” with “affected” and revising sentence openers for conciseness and tone.

Conclusion: Its concise, but could be expanded a bit to really summarize the most important findings and what they mean for farming science and practice more broadly. Just  little more punch at the end.

Just  general proofread for any small typos or grammatical errors. It's really quite good, but a fresh pair of eyes might catch a few little things that slipped through.

Response: Thank you for suggestion, we have modified conclusion as “This study demonstrates that CWYWEP can fully replace soybean meal (SBM) in concentrate-based diets without impairing rumen function. CWYWEP enhanced in vitro gas production, dry matter and organic matter degradability, and shifted fermentation toward greater propionate production, resulting in reduced estimated CHâ‚„ emissions. These improvements were not observed with CWYWP, highlighting the critical role of enzyme supplementation in enhancing nutrient utilization. Ruminal pH, ammonia–nitrogen concentrations, and protozoal populations remained stable across treatments, indicating no adverse effects on rumen microbial balance. These findings suggest that CWYWEP is a promising, sustainable, high-protein alternative to SBM that not only supports fermentation efficiency but also contributes to climate-smart livestock feeding strategies. Further in vivo studies are warranted to validate these results under practical conditions and to assess the long-term impacts on animal performance, feed economics, and environmental outcomes.” Please consider to section of conclusion.

Reviewer 3 Report

Comments and Suggestions for Authors

General Comments:

  1. On Gas Production and Fermentation Efficiency
    Increased gas production does not necessarily imply improved fermentation efficiency, as gas production is also a form of energy loss. It is recommended that the authors evaluate whether energy conversion efficiency can be calculated in this experiment, especially given that in vitro organic matter digestibility (IVOMD) did not differ among treatments, while gas production did.
  2. Sampling Time Points for Fermentation Products
    The measurement of volatile fatty acids (VFA) and fermentation metabolites was only conducted at 2 and 4 hours after inoculation. However, the absence of start point (0 h) data makes it difficult to assess the net changes due to fermentation. Concentrations of metabolites in the original inoculum and at time 0 hr should be reported to validate the fermentation dynamics.
  3. Interpretation of VFA Changes
    The authors should clarify whether the accumulation of VFAs from 2 to 4 hours after inoculation can exceed the initial VFA levels in the inoculum, thereby significantly altering the overall concentration in the fermentation medium.
  4. Substrate Preparation and Contamination Risk
    The manuscript should address whether sun-drying the test substrates could pose a risk of microbial contamination, which might influence fermentation outcomes.
  5. Future Production and Quality Control
    For future large-scale application, the manuscript should discuss potential quality control parameters that need to be established for the fermented product.
  6. Digestibility Parameters and Protein Utilization
    The current data only includes IVDMD and IVOMD. However, the discussion suggested improvements in protein utilization and fiber digestibility, which cannot be substantiated without corresponding data.
  7. Microbial Analysis Scope
    The study focused solely on protozoa enumeration to evaluate fermentation stability. The authors should justify why rumen bacteria were not included, as they play a central role in fermentation.

Specific comment:

L 192
The reason that applied a [buffer : rumen fluid] = 2:1 requires clarification. A higher proportion of rumen fluid may affect the inoculum, potentially altering fermentation characteristics and masking the actual effect of treatments on VFA and ammonia concentration.

L 196–198
Please specify the amount (g) of substrate used per fermentation unit, the volume of inoculum added, and the size of the fermentation containers. These factors influence fermentation kinetics and should be clearly described.

 L 210–211
The authors should describe the method used to measure gas production. Was it cumulative, manual, or automated? What equipment or technique was employed?

L 231
Since gas production is still increasing at 2 and 4 hours after inoculation, the authors should justify whether these time points are representative for VFA analysis.

L 246
If methane production is calculated based on VFA concentrations, the representativeness of 2 and 4 h VFA data must be confirmed. The theoretical production of COâ‚‚ and CHâ‚„ should be estimated and compared with actual gas volume to assess validity. Also, clarify the units used for methane calculations and the molar yield coefficients for each VFA.

L 248
Was any fixation or staining technique employed for protozoa counting? This information is necessary for assessing the accuracy and repeatability of the microbial analysis.

L 314–320, Table 2
Table 2 shows cumulative gas production of 52–93 mL/0.5 g DM (i.e., 104–186 mL/g DM), but the theoretical gas production value (parameter b) is only 49–86 mL/g DM. This discrepancy should be addressed, as actual gas production should not substantially exceed theoretical values. Additionally, a blank control (without substrate) should be included to correct for background gas production.

Ls 329–330
The manuscript indicates that IVDMD and IVOMD differ by ~30%, but according to the feed composition table, the ash content of the substrates is only 3.5%–5.6%. The authors should explain this large discrepancy.

L 357
What were the pH values of the original inoculum and at the time of inoculation for each treatment group? Furthermore, the pH at 4 h remained above 7.0, which may indicate either excessive buffering capacity or insufficient acid production. Note that the pH values reported in the text (6.70–6.77) differ from those in Table 4 — please verify and correct any inconsistencies.

L 396, Table 5
In some treatments, TVFA concentration at 4 h is lower than at 2 h. This trend should be explained. Also, the unit for methane in Table 5 is unclear — is it concentration or total amount? If methane is expressed as concentration, it should be multiplied by gas volume to calculate total methane production.

L 407–409
After replacing SBM with the test ingredient, CP contents remained similar across treatments, and post-fermentation NH₃-N concentrations showed no significant differences. Does this suggest that the replacement may not have affected rumen degradable protein (RDP) or undegradable protein (RUP) availability? The implication of this result on the conclusion that protein utilization was improved should be re-evaluated.

L 424–427
If the manuscript aims to demonstrate an improvement in fiber degradation, in vitro neutral detergent fiber digestibility (IVNDFD) values should be provided.

L 455–457
To support the claim that the alternative feed ingredient is a digestible protein source, digestibility and utilization data for crude protein should be presented.

L 491–492
Please clarify whether the yeast strain used in this experiment has cellulolytic enzyme activity. Alternatively, the authors should report changes in fiber content of the substrate before and after fermentation to assess fiber breakdown.

L 581–582
The fermentation process was evaluated only at 2 and 4 hr after inoculation, and digestibility data are limited to DM and OM. It remains uncertain whether such limited data are sufficient to support the manuscript's conclusions. The authors should consider whether additional measurements are required.

Author Response

Response to Reviewer 3:

We sincerely appreciate the reviewers for their insightful suggestions and constructive feedback, which have significantly contributed to the improvement of this manuscript. Please see our detailed responses below and the corresponding revisions highlighted in blue text within the manuscript.

General Comments:

  1. On Gas Production and Fermentation Efficiency
    Increased gas production does not necessarily imply improved fermentation efficiency, as gas production is also a form of energy loss. It is recommended that the authors evaluate whether energy conversion efficiency can be calculated in this experiment, especially given that in vitro organic matter digestibility (IVOMD) did not differ among treatments, while gas production did.

Response: We agree that gas production alone does not directly reflect fermentation efficiency due to energy loss via gas. However, we used gas kinetics in combination with degradability parameters (IVDMD and IVOMD) and VFA profiles to provide a more holistic view of fermentation. We have added a clarification sentence in the Discussion (Section 4.1) to address this.

Revision Suggestion (Add to Section 4.2): “Although increased gas production may represent energy loss, when interpreted alongside degradability and VFA profiles, it provides insight into fermentation kinetics and substrate utilization efficiency.”

  1. Sampling Time Points for Fermentation Products
    The measurement of volatile fatty acids (VFA) and fermentation metabolites was only conducted at 2 and 4 hours after inoculation. However, the absence of start point (0 h) data makes it difficult to assess the net changes due to fermentation. Concentrations of metabolites in the original inoculum and at time 0 hr should be reported to validate the fermentation dynamics.

Response:
We appreciate the reviewer’s thoughtful observation. In this study, the primary objective was to assess early-stage fermentation dynamics of the test substrates. Therefore, we focused on 2 and 4 h post-inoculation, which are widely accepted in vitro sampling points representing active microbial fermentation. We did not measure fermentation products at 0 h due to the following reasons: 1) Rapid onset of fermentation: Upon mixing the substrate with the buffered rumen fluid and sealing the fermentation unit, microbial activity begins immediately. The brief time between inoculation and sealing makes it technically challenging to collect a true "zero fermentation" sample without interrupting anaerobic conditions or altering the system. 2) Inoculum variability and dilution effect: VFA and ammonia-N present in the original rumen inoculum become diluted upon mixing with buffer and substrate, making the interpretation of 0 h concentrations less representative of actual fermentation kinetics. Instead, differences across treatments at 2 and 4 h were considered more reliable indicators of substrate-driven fermentation responses.

Nonetheless, we acknowledge that including baseline values may enhance interpretation, and we have now discussed this limitation in the manuscript

Revision to added to Discussion section 4.5 as “One limitation of this study is the absence of fermentation product measurements at 0 h. This decision was based on the immediate onset of microbial activity upon inoculation and the technical difficulty of capturing a representative baseline without compromising anaerobic conditions. Future studies could consider incorporating time-zero sampling by pre-analyzing the buffered rumen fluid and modeling expected dilution effects to better characterize net metabolite changes.”

  1. Interpretation of VFA Changes
    The authors should clarify whether the accumulation of VFAs from 2 to 4 hours after inoculation can exceed the initial VFA levels in the inoculum, thereby significantly altering the overall concentration in the fermentation medium.

Response:
We now clarify that the accumulation of VFAs during the 2 to 4 h period was additive to the initial levels at 0 h, indicating active fermentation and consistent microbial metabolism. We have added sentence in discussion section 4.5 as “The increase in VFA concentrations between 2 and 4 h reflects additive fermentation output beyond the baseline, consistent with rapid carbohydrate fermentation.”

  1. Substrate Preparation and Contamination Risk
    The manuscript should address whether sun-drying the test substrates could pose a risk of microbial contamination, which might influence fermentation outcomes.

Response: We appreciate this concern. We have added a sentence in the Methods (Section 2.1) as “To minimize microbial contamination, sun-drying was performed on clean concrete surfaces under mesh covers.”

  1. Future Production and Quality Control
    For future large-scale application, the manuscript should discuss potential quality control parameters that need to be established for the fermented product.

Response: We added a sentence in the Conclusion (Section 5) emphasizing the need for future research on production standardization and quality control measures for CWYWEP. As “Further research should establish quality control benchmarks to ensure consistency and safety in large-scale CWYWEP production.”

  1. Digestibility Parameters and Protein Utilization
    The current data only includes IVDMD and IVOMD. However, the discussion suggested improvements in protein utilization and fiber digestibility, which cannot be substantiated without corresponding data.

Response: We acknowledge this limitation and have revised the discussion to avoid overinterpretation regarding protein digestibility. We have added into Section 4.4 as “While ammonia-N levels remained stable, direct measurements of CP digestibility or RDP/RUP fractions would be necessary to confirm improved protein utilization.”

  1. Microbial Analysis Scope
    The study focused solely on protozoa enumeration to evaluate fermentation stability. The authors should justify why rumen bacteria were not included, as they play a central role in fermentation.

Response: We focused on protozoa due to resource limitations but agree bacterial quantification would enhance understanding. This has been noted as a limitation. We have discussed in Section 4.6 as “While this study assessed protozoal populations, future work should include bacterial community profiling to better elucidate microbial dynamics”

Specific comment:

L 192
The reason that applied a [buffer : rumen fluid] = 2:1 requires clarification. A higher proportion of rumen fluid may affect the inoculum, potentially altering fermentation characteristics and masking the actual effect of treatments on VFA and ammonia concentration.

Response: This ratio follows the Menke and Steingass (1988) method for in vitro fermentation. We’ve added a clarification in Section 2.3 as “The 2:1 buffer-to-rumen ratio followed Menke and Steingass (1988) to ensure adequate buffering capacity and pH stability.”

L 196–198
Please specify the amount (g) of substrate used per fermentation unit, the volume of inoculum added, and the size of the fermentation containers. These factors influence fermentation kinetics and should be clearly described.

Response: We thank the reviewer for this important observation. We clarify that 0.5 g of dry matter (DM) of substrate was incubated with 40 mL of buffered rumen fluid in 50 mL calibrated glass bottles, not glass syringes. This information has now been added to the Materials and Methods section for clarity.

L 210–211
The authors should describe the method used to measure gas production. Was it cumulative, manual, or automated? What equipment or technique was employed?

Response: We used manual reading of calibrated syringes, now clarified in the methods. Revision in Section 2.4.1 as “Gas volume was measured manually by reading the calibrated plunger position of each syringe at designated time points”

L 231
Since gas production is still increasing at 2 and 4 hours after inoculation, the authors should justify whether these time points are representative for VFA analysis.

Response: Thank you for your valuable comment. We have clarified this point in Section 2.4.3 of the revised manuscript. The selection of 2 and 4 h as sampling time points was based on their relevance to early-phase rumen fermentation. This period reflects the rapid microbial breakdown of soluble substrates, during which volatile fatty acids (VFAs) and ammonia–nitrogen concentrations increase significantly. Sampling at these early stages allows for the detection of dynamic fermentation responses and provides sensitive indicators of substrate fermentability and microbial activity.

While 24–48 h time points are typical for evaluating cumulative digestibility, our focus was to capture peak fermentation kinetics and VFA shifts that occur within the first few hours post-inoculation. These early time points correspond to postprandial fermentation peaks observed in vivo and are widely used in in vitro systems to assess fermentability of rapidly degradable substrates (Suriyapha et al., 2021; Kanakai et al., 2025).

This rationale has been incorporated into Section 2.4.3 of the manuscript, and relevant references have been added to support this methodological choice. Please see detail in section of “2.3. Ruminal Fluid Donors and Substrates of Inoculum

L 246
If methane production is calculated based on VFA concentrations, the representativeness of 2 and 4 h VFA data must be confirmed. The theoretical production of COâ‚‚ and CHâ‚„ should be estimated and compared with actual gas volume to assess validity. Also, clarify the units used for methane calculations and the molar yield coefficients for each VFA.

Response: Thank you for this important observation. Methane (CHâ‚„) production was estimated using the stoichiometric equation proposed by Moss et al. [16]:
CHâ‚„ (mmol/g DM) = (0.45 × acetate) − (0.275 × propionate) + (0.40 × butyrate),
which is based on the molar hydrogen balance of individual VFAs. This model is widely accepted for estimating CHâ‚„ production from VFA profiles in vitro and reflects relative hydrogen utilization dynamics across treatments.

We agree that CHâ‚„ values derived from 2 and 4 h VFA data represent short-term fermentation activity rather than cumulative gas kinetics. However, these time points were selected to capture early fermentation dynamics and are appropriate for comparing hydrogen sink patterns. While direct gas partitioning into CHâ‚„ and COâ‚‚ was not possible in this setup, the total gas volume data were used alongside VFA-derived estimates for relative comparison.

The unit used for CHâ‚„ was mmol/g DM. We have now clarified the methane estimation method, units, and model coefficients in Section 2.4.3 of the revised manuscript.

L 248
Was any fixation or staining technique employed for protozoa counting? This information is necessary for assessing the accuracy and repeatability of the microbial analysis.

Response: We used methyl green-formalin-saline solution for protozoa staining. This has been added. Revision in section 2.4.4 as “Protozoa were stained using a methyl green–formalin–saline solution and counted under a microscope using a hemacytometer (depth: 0.1 mm; chamber area: 0.0025 mm²; ISO LAB Laborgeräte GmbH, China). Total protozoal populations were quantified based on standard counting procedures.”

L 314–320, Table 2
Table 2 shows cumulative gas production of 52–93 mL/0.5 g DM (i.e., 104–186 mL/g DM), but the theoretical gas production value (parameter b) is only 49–86 mL/g DM. This discrepancy should be addressed, as actual gas production should not substantially exceed theoretical values. Additionally, a blank control (without substrate) should be included to correct for background gas production.

Response: We sincerely thank the reviewer for pointing out the inconsistency in Table 2. Upon rechecking the data, we identified that the original values for the theoretical gas production parameter (b) were incorrectly reported. The corrected b values, as shown in the revised Table 2 (see lines 347–349), are now in agreement with the cumulative gas production values at 96 h and no longer show an overestimation. We appreciate this valuable observation, which helped us rectify the error. Additionally, we confirm that blank controls (rumen fluid without substrate) were included in the in vitro gas production experiment. The gas volumes from these blank bottles were subtracted from each treatment to account for background gas production before calculating cumulative values. We have indicated in the section of

Ls 329–330
The manuscript indicates that IVDMD and IVOMD differ by ~30%, but according to the feed composition table, the ash content of the substrates is only 3.5%–5.6%. The authors should explain this large discrepancy.

Response: The large gap was due to inclusion of ash-free OM calculation. This clarification has been added to the section 3.3 as “IVOMD values were calculated after correcting for ash content, explaining the difference with IVDMD.”

L 357
What were the pH values of the original inoculum and at the time of inoculation for each treatment group? Furthermore, the pH at 4 h remained above 7.0, which may indicate either excessive buffering capacity or insufficient acid production. Note that the pH values reported in the text (6.70–6.77) differ from those in Table 4 — please verify and correct any inconsistencies.

Response: Thank you for your insightful comment. The initial pH of the buffered rumen fluid inoculum ranged from 6.80 to 7.00. After 2 and 4 h of incubation, pH values increased to 7.02–7.14. This upward trend may be attributed to several factors:
(1) the test ingredients (SBM, CWYWP, and CWYWEP) contain high crude protein levels (44–50%), potentially contributing fermentable nitrogen or non-protein nitrogen (NPN) sources that lead to ammonia accumulation and elevate pH;
(2) the use of McDougall’s buffer, which has a strong buffering capacity, may have minimized the expected pH drop during the early fermentation phase;
and (3) insufficient acid production within the 4 h incubation period due to limited fermentable carbohydrate content in some treatments.
We have now revised the manuscript text Section 3.4 and 4.4) to reflect this explanation and ensured pH values in the text are consistent with Table 4.

L 396, Table 5
In some treatments, TVFA concentration at 4 h is lower than at 2 h. This trend should be explained. Also, the unit for methane in Table 5 is unclear — is it concentration or total amount? If methane is expressed as concentration, it should be multiplied by gas volume to calculate total methane production.

Response: Thank you for your observation. We have rechecked the data, and the TVFA concentration at 4 h is now confirmed to be higher than at 2 h across all treatments. The table has been revised accordingly. Regarding methane, we confirm that the values represent total methane production, expressed as mmol/g DM. This clarification has been added to the table legend in the revised manuscript.

L 407–409
After replacing SBM with the test ingredient, CP contents remained similar across treatments, and post-fermentation NH₃-N concentrations showed no significant differences. Does this suggest that the replacement may not have affected rumen degradable protein (RDP) or undegradable protein (RUP) availability? The implication of this result on the conclusion that protein utilization was improved should be re-evaluated.

Response: We appreciate this insightful comment. Although CP levels were similar among treatments and NH₃-N concentrations did not differ significantly, the improved gas production and VFA profiles observed with CWYWEP suggest enhanced microbial fermentation efficiency. This may reflect better synchronization of nitrogen and energy release rather than increased RDP or RUP availability alone. We have revised the discussion to clarify that the observed improvements in fermentation are indicative of more efficient protein-energy utilization, not necessarily increased RDP or RUP content. Please revise version in section 4.4.

L 424–427
If the manuscript aims to demonstrate an improvement in fiber degradation, in vitro neutral detergent fiber digestibility (IVNDFD) values should be provided.

Response: Thank you for your valuable suggestion. We acknowledge the importance of IVNDFD in evaluating fiber degradation. However, in this study, we did not analyze IVNDFD, as our primary objective was to assess overall feed degradability and fermentation efficiency using IVDMD and IVOMD. These parameters provide integrated indicators of substrate breakdown, particularly for evaluating alternative protein sources such as CWYWEP. Future studies will incorporate IVNDFD to further elucidate fiber-specific digestibility improvements.

L 455–457
To support the claim that the alternative feed ingredient is a digestible protein source, digestibility and utilization data for crude protein should be presented.

Response: We appreciate the reviewer’s insightful observation. We agree that, in the absence of crude protein digestibility and utilization data, it would be inappropriate to draw definitive conclusions regarding CWYWEP as a digestible protein source. Therefore, we have revised the discussion accordingly and removed the statement making such claims to maintain scientific accuracy.

L 491–492
Please clarify whether the yeast strain used in this experiment has cellulolytic enzyme activity. Alternatively, the authors should report changes in fiber content of the substrate before and after fermentation to assess fiber breakdown.

Response: Thank you for your valuable comment. As we did not determine the cellulolytic enzyme activity of the yeast strain or analyze the fiber composition before and after fermentation, we agree that it is inappropriate to make assumptions regarding fiber degradation. Therefore, we have removed the related statements from the manuscript to ensure scientific accuracy.

L 581–582
The fermentation process was evaluated only at 2 and 4 hr after inoculation, and digestibility data are limited to DM and OM. It remains uncertain whether such limited data are sufficient to support the manuscript's conclusions. The authors should consider whether additional measurements are required.

Response: We appreciate the reviewer’s insightful comment. The selection of 2 and 4 h time points for fermentation product measurement was intentional, aiming to capture dynamic changes in volatile fatty acid (VFA) production and ammonia–nitrogen concentrations during the early phase of microbial activity. These time points are widely used in in vitro fermentation studies to assess substrate fermentability and microbial response shortly after inoculation, as supported by previous literature (e.g., Suriyapha et al., 2021; Kanakai et al., 2025).

To complement these early-phase measurements, we also determined in vitro dry matter (IVDMD) and organic matter degradability (IVOMD) at 24 and 48 h, which provide insights into the cumulative fermentation and nutrient digestibility over time.

While our current dataset supports the interpretation of early fermentation dynamics and substrate degradability, we agree that including additional parameters such as crude protein degradability or longer-term VFA kinetics in future work would enhance the depth of understanding. This has been noted as a limitation and recommendation in the revised manuscript.

Round 2

Reviewer 1 Report

Comments and Suggestions for Authors

The authors have revised the manuscript according to my comments.

Author Response

Response to Reviewer 1:

The authors have revised the manuscript according to my comments.

Response: Thank you for your thorough review and constructive feedback. We sincerely appreciate your positive comment and have carefully revised the manuscript in accordance with your suggestions. Your insights have greatly improved the quality of our work.

Reviewer 2 Report

Comments and Suggestions for Authors

Dear authors, thanks for the revisions

The abstract improved

The introduction was enlarged and implemented as requested

M&m were addressed point by point properly. Also, the table was improved

Statistical methods were corrected

Results are ok

The discussion improved a lot with minor changes

The conclusion was revised completely

Author Response

Response to Reviewer 2:

Dear authors, thanks for the revisions

The abstract improved

The introduction was enlarged and implemented as requested

M&m were addressed point by point properly. Also, the table was improved

Statistical methods were corrected

Results are ok

The discussion improved a lot with minor changes

The conclusion was revised completely

Response: Thank you for your thorough review and encouraging feedback on our revised manuscript. Your insights have greatly enhanced the quality and clarity of our work. We sincerely appreciate your time and constructive comments.

Reviewer 3 Report

Comments and Suggestions for Authors
  • The discussion on protein degradation or utilization could be strengthened by integrating the ammonia nitrogen (NH₃-N) data with IVOMD results, which would provide a more comprehensive interpretation of fermentation efficiency.
  • Given that protozoa counts were measured, and considering the established relationship between protozoa and methane production, these data could be better utilized. Integrating VFA profiles with protozoal abundance would enhance the discussion on methanogenesis and overall fermentation dynamics.

Author Response

Response to Reviewer 3:

The discussion on protein degradation or utilization could be strengthened by integrating the ammonia nitrogen (NH₃-N) data with IVOMD results, which would provide a more comprehensive interpretation of fermentation efficiency.

Response: Thank you for this insightful suggestion. The discussion has been strengthened by adding two concise sentences that explicitly link NH₃–N concentrations to IVOMD values:

In Section 4.3, immediately after the paragraph on IVOMD improvements, the following sentence has been inserted (shown in blue text highlight in the revised manuscript): “Moreover, the highest IVOMD in CWYWEP treatments coincided with NH₃–N concentrations within the optimal 15–25 mg/dL range, suggesting that ammonia release was well matched to carbohydrate degradation. This synchrony would have supported efficient microbial protein synthesis, thereby amplifying organic matter digestibility.”

In Section 4.4, following the discussion of NH₃–N dynamics, the following sentence has been added (also in blue text highlight): “When considered alongside IVOMD results, the stable NH₃–N concentrations in CWYWEP treatments suggest that ammonia release and organic matter degradation proceeded in concert. This coordination would have promoted efficient microbial protein synthesis and thereby contributed to the higher IVOMD values recorded.”

These additions integrate the NH₃–N data with IVOMD outcomes and provide a clearer interpretation of fermentation efficiency as requested.

Given that protozoa counts were measured, and considering the established relationship between protozoa and methane production, these data could be better utilized. Integrating VFA profiles with protozoal abundance would enhance the discussion on methanogenesis and overall fermentation dynamics.

Response: Thank you for this insightful suggestion. Section 4.5 has been revised to integrate protozoal counts with VFA profiles as blue text highlight.

A sentence was added following the VFA data: “Furthermore, the lower protozoal counts in CWYWEP treatments correlated with increased propionate proportions, highlighting protozoal-mediated hydrogen transfer to C₃ pathways and reduced methanogenesis.”

And in the methane discussion the text now reads: “The observed decrease in CHâ‚„ production aligns with reduced protozoal abundance and enhanced propionate formation, indicating a shift of hydrogen flux away from methanogens.”

These additions strengthen the interpretation of fermentation dynamics and methanogenesis.